# Pseudovirus-Based Systems for Screening Natural Antiviral Agents: A Comprehensive Review

**DOI:** 10.3390/ijms25105188

**Published:** 2024-05-10

**Authors:** Paola Trischitta, Maria Pia Tamburello, Assunta Venuti, Rosamaria Pennisi

**Affiliations:** 1Department of Chemical, Biological, Pharmaceutical and Environmental Science, University of Messina, Viale Ferdinando Stagno d’Alcontres 31, 98166 Messina, Italy; paola.trischitta@studenti.unime.it (P.T.); maria.tamburello1@studenti.unime.it (M.P.T.); 2Department of Chemistry, Biology, and Biotechnology, University of Perugia, Via Elce di Sotto 8, 06123 Perugia, Italy; 3International Agency for Research on Cancer (IARC), World Health Organization, 69366 Lyon, CEDEX 07, France; assuntavenuti@gmail.com

**Keywords:** pseudovirus, natural products, antivirals, emerging viruses

## Abstract

Since the outbreak of COVID-19, researchers have been working tirelessly to discover effective ways to combat coronavirus infection. The use of computational drug repurposing methods and molecular docking has been instrumental in identifying compounds that have the potential to disrupt the binding between the spike glycoprotein of SARS-CoV-2 and human ACE2 (hACE2). Moreover, the pseudovirus approach has emerged as a robust technique for investigating the mechanism of virus attachment to cellular receptors and for screening targeted small molecule drugs. Pseudoviruses are viral particles containing envelope proteins, which mediate the virus’s entry with the same efficiency as that of live viruses but lacking pathogenic genes. Therefore, they represent a safe alternative to screen potential drugs inhibiting viral entry, especially for highly pathogenic enveloped viruses. In this review, we have compiled a list of antiviral plant extracts and natural products that have been extensively studied against enveloped emerging and re-emerging viruses by pseudovirus technology. The review is organized into three parts: (1) construction of pseudoviruses based on different packaging systems and applications; (2) knowledge of emerging and re-emerging viruses; (3) natural products active against pseudovirus-mediated entry. One of the most crucial stages in the life cycle of a virus is its penetration into host cells. Therefore, the discovery of viral entry inhibitors represents a promising therapeutic option in fighting against emerging viruses.

## 1. Introduction

Pseudoviruses are replication-defective viruses able to enter susceptible host cells as well as authentic viruses due to the conformational structures of viral glycoproteins, highly similar to native glycoproteins [1,2,3]. Due to their high pathogenicity and infectivity, some viruses must be handled under biosafety level 3 conditions. Unlike authentic viruses, pseudoviruses lack pathogenic genes, have no autonomous replication ability, and can replicate intracellularly for a single cycle only [2,4]. Therefore, pseudoviruses can be safely manipulated in biosafety level 2 facilities [3,5] to study the internalization of highly infectious viruses and other aspects of viral infectivity, such as cellular tropism [6], and/ or the identification of cellular receptors without the need for high-level containment measures [7,8]. In addition, they can be used for developing, screening, and evaluating monoclonal antibodies, vaccines [9,10], and antiviral drugs [11], as well as for detecting neutralizing antibody titers in serum after vaccination [5]. In this review, we describe the construction of pseudoviruses based on different packaging systems and their current applications, focusing on the usage of pseudoviruses for screening natural antiviral drugs [8]. 

Indeed, many studies have shown that a wide variety of active phytochemicals influence cellular functions, membrane permeability, and viral replication [12]. Thus, natural product-based drug development may be useful for treating viral diseases [13,14,15]. Antiviral drugs can be classified according to their chemical nature or mechanisms of action against viral or cellular host proteins. They can inhibit viral entry, viral RNA synthesis, and viral replication [16,17,18]. This review collects a series of antiviral compounds that have been tested and whose mechanism of action has been identified using pseudovirus technology.

## 2. Packaging Systems and Strategies for Pseudovirus Production 

The assembly of pseudoviruses relies on three distinct packaging systems: (1) the human immunodeficiency virus-1 (HIV-1)-based lentiviral packaging system; (2) the murine leukemia virus (MLV)-based packaging system; and (3) the vesicular stomatitis virus (VSV) packaging system (Figure 1). Several factors, such as timing efficiency, established protocols, production challenges, and yield limitations, must be considered when choosing a pseudovirus packaging system. HIV and MLV packaging systems are easier and less time-consuming compared to the VSV packaging method. Nevertheless, VSV systems typically yield higher pseudovirus quantities, and HIV and MLV systems might yield comparatively lower quantities, impacting experimental scalability and outcomes. In addition, VSV and MLV systems are safer than HIV systems. In both cases, the pseudoviruses’ surface will bear envelope proteins derived from the viruses under investigation.

### 2.1. The Human Immunodeficiency Virus (HIV-1)-Based Lentiviral Packaging System

The HIV pseudovirus system is widely utilized due to its versatility and well-established methodology [19,20]. The HIV genome comprises structural, regulatory, and helper genes, carefully manipulated to construct pseudoviruses. To reduce undesired recombination events, distinct segments of the HIV genome are cloned into different DNA expression vectors. Moreover, some dispensable elements are removed during the cloning process. Additionally, mutations in the envelope gene allow for the incorporation of envelope proteins from other viruses [21]. Depending on the system’s complexity, HIV pseudovirus construction can involve two, three, or four plasmids. 

The commonly used two-plasmid system includes an expression plasmid and a packaging plasmid, such as pSG3Δenv or pNL4–3 [22]. The three-plasmid system separates HIV components into packaging and transfer plasmids, while the four-plasmid system further divides the components to enhance safety. The transfer and packaging plasmids that are most frequently utilized are psPAX2 and pLenti-GFP [23,24].

The transfection of these plasmids into host cells results in the assembly of incomplete HIV pseudovirus particles, which are secreted into the extracellular space. During secretion, heterologously expressed envelope proteins are integrated into the viral membrane. The subsequent step of collection, centrifugation, purification, and concentration yields pseudovirus particles suitable for subsequent research [2]. The generated virus-like particles could mediate a particular viral life cycle but are faulty in replication and lack protein-coding genes [25]. The third generation of the packaging system is designed with numerous safety precautions: the transferred plasmids must be replication-incompetent and become “self-inactivating” after integration by shortened 3′ long terminal repeat LTR. Nevertheless, there is still a potential risk of tumor gene activation due to the insertion of transgene sequences into the host genome via viral transduction [26]. Lentiviral pseudovirus systems have been developed for SARS-CoV, SARS-CoV-2, and avian influenza H5 [27]. An envelope expression plasmid, derived from the pMD.G vector, was used for cloning the genes that encode structural proteins, such as the S protein of SARS-CoV or SARS-CoV-2, and HA/NA proteins of avian influenza H5. To construct these pseudoviruses, the plasmids carrying the structural protein genes (S or HA and NA proteins), a packaging vector, and a reporter vector were simultaneously transfected into HEK-293T cells. The characterization of pseudoviral particles was carried out by verifying the expression of the reporter gene through transduction into Vero-E6 or MDCK cells. Additionally, lentivirus-based pseudotypes were developed to study Ebola virus entry and bear GP glycoprotein on the surface, as detailed by Cao and coauthors [28]. The characterization of pseudotyped EBOV included electron microscopy of pseudo-EBOV and the quantification of relative light units (RLUs). The highest titer of pseudotyped EBOV was observed between 24 to 36 h post-transfection. This result confirmed the single-cycle infectivity of pseudotyped EBOV and its ability to infect target cells effectively within a specific timeframe after transfection. Additionally, the authors also reported cell susceptibility of pseudo-EBOV in different cell lines. This represents an integral part of the production and characterization strategies employed for pseudoviruses. Similarly, a Nipah pseudovirus (NiV) bearing NiV-F and -G proteins on the particle surface was developed for in vivo infection to mimic infectious NiV for the entry process [29]. A Nipah pseudovirus mouse model bearing NiV-F and -G proteins on the particle surface was developed to mimic the entry process of NiV in vivo infection. Balb/c mice were inoculated with pseudotyped virus by the intrathoracic route, and a luminescence detector reported high flux levels in the spleen and the lung. NiV pseudovirus was enrolled for neutralization assays in vivo which were reported to be less labor-intensive and more rapid than traditional assays. 

### 2.2. The Murine Leukemia Virus (MLV)-Based Packaging System

MLV is a retrovirus similar to HIV, primarily infecting mice, and associated with leukemia. MLV possesses three structural genes (gag, pol, and env) that, respectively, encode for viral capsid and nucleocapsid proteins, viral enzymes (reverse transcriptase, integrase, and protease), and envelope proteins [30]. By cloning two gene sets into plasmids, one containing gag-pol genes and the other harboring the reporter gene, it is possible to obtain a highly effective MLV packaging method [31]. The process of packaging of MLV pseudoviruses mirrors that of HIV. Plasmid transfection is used to introduce the gag and pol structural genes and a gene that encodes a heterologous virus envelope protein into the cells. After intracellular recombination, pseudovirus particles that bear the heterologous virus envelope proteins are released into the medium of the cell culture. The “three-plasmid” co-transfection method, reported in reference [32], involves: (i) a plasmid containing the packaging construct that facilitates the expression of MLV retroviral core genes, gag, and pol, but without the MLV envelope glycoprotein-encoding env gene; (ii) a minimal retroviral transfer vector that carries the luciferase reporter gene, retroviral regulatory LTR regions, and a packaging signal; (iii) and an expression construct that encodes the desired glycoprotein [1,5,33]. The simultaneous expression of these three plasmids enables the production of LTR-flanked reporter gene-containing RNA, MLV-derived proteins, and heterologous envelope glycoprotein. RNAs carrying the LTR-flanked luciferase gene are integrated into the developing particles created by the assembly and budding of MLV capsid proteins. This process also attracts heterologous viral envelope glycoproteins during pseudotyped particle production at the plasma membrane. An MLV-based pseudovirus was also employed to study SARS-CoV-2 entry and to test potential inhibitors for their ability to block viral internalization. For example, the SARS-CoV-2 MLV pseudovirus was packaged using a co-transfection method with three plasmids, resulting in a high yield. The plasmids included a packaging plasmid (SV-Psi-Env-MLV), a luciferase reporter plasmid with regulatory elements (L-LUC-SN), and an expression plasmid for SARS-CoV-2 proteins [34].

Pseudotyped particles are safer, since they permit only one round of infection. This is because the integrated sequence carries only the luciferase reporter gene and none of the MLV genes. After the transduction, the infectivity of the pseudotyped particles can be measured by detecting and quantifying the expression of the luciferase reporter gene [33]. In some cases, it was reported that prolonged incubation periods exceeding 24 h resulted only in minimal increases in the detected luminescence intensity [34]. 

The murine leukemia virus (MLV) represents a more advantageous packaging system than HIV. For instance, studies conducted by Cosset and collaborators demonstrated that the MLV system exhibited five times greater efficiency than the HIV system when Lassa virus-mediated entry into cells was evaluated. This highlights the potential superiority of the MLV system in facilitating viral entry studies and related investigations [35,36]. 

### 2.3. The VSV Packaging System

The VSV packaging system stands out as an ideal tool for generating pseudotyped viruses due to several advantageous characteristics. First, its simple genome structure facilitates manipulation. Secondly, VSV exhibits the ability to infect a wide variety of animal cells, enhancing its versatility. Thirdly, it lacks stringent selectivity for envelope proteins, allowing for flexibility in experimental designs. Moreover, the capability to work with VSV in a BSL-2 laboratory further enhances its utility and accessibility for the researchers. The genome of VSV, 11 kb negative single-strand RNA (ssRNA), encodes for five major proteins: nuclear (N), phosphate (P), matrix (M), glycoprotein (G), and RNA polymerase (L) [37,38]. The glycoprotein (G protein) of VSV plays a crucial role in facilitating the attachment and the entry of VSV into susceptible host cells, thus contributing significantly to virus infectivity. A reverse genetics approach was used to remove G protein from VSV, resulting in a modified VSV termed rVSV-ΔG [39]. This engineered rVSV-ΔG serves as a versatile platform for producing VSV pseudotypes harboring envelope glycoproteins from various heterologous viruses, including those necessitating high-level containment such as encephalitis viruses, SARS-CoV, SARS-CoV-2, and MERS coronaviruses. 

## 3. Pseudovirus Applications

Pseudoviruses are engineered particles that mimic the structure of authentic viruses but cannot replicate. They have become invaluable tools in various fields of virology and medical research. Understanding viral entry mechanisms, vaccine development, antiviral drug screening, and virus neutralization assays are just some of the applications they can be applied to.

### 3.1. Mechanisms of Virus Entry

Studying viral entry mechanisms is crucial for unraveling the pathogenesis of viral infections. Pseudoviruses provide a controlled system to investigate the intricate processes involved in viral attachment, fusion, and internalization and gain insights into viral tropism, host cell receptors, and immune evasion strategies. 

Pseudotyped viruses, such as those based on HIV, have played a pivotal role in uncovering key insights into viral pathogenesis and host–virus interactions. For instance, studies utilizing pseudotyped HIV have been instrumental in identifying critical factors in HIV-1 infection. One landmark discovery was the identification of CCR5 as a major coreceptor for primary isolates of HIV-1, shedding light on the mechanisms of viral entry [40]. Additionally, using pseudotyped HIV, we have learned about the role of TRIM5α, a cytoplasmic protein that can inhibit HIV-1 infection in Old World monkeys. This has provided crucial insights into host defense mechanisms against retroviral infections [41]. Similarly, pseudotyped HCV (hepatitis C virus) particles have been employed to investigate the entry mechanisms of HCV into host cells. These studies have contributed to the identification of HCV receptors and the characterization of pH-dependent cell entry mediated by HCV glycoproteins [42]. Moreover, the pseudovirus construction was useful for establishing the cell tropism of MERS-CoV. A pseudovirus with the full-length spike protein (S) from MERS-CoV was created using an Env-defective HIV-1 backbone that expressed luciferase as a reporter gene. Looking at viral internalization in both human and non-human hosts, it was discovered that MERS-CoV pseudovirus enabled single-cycle infection in different cell types expressing dipeptidyl peptidase-4 (DPP4), which was confirmed as the receptor for MERS-CoV [19]. During the SARS-CoV-2 pandemic, the use of pseudoviruses led to several significant advancements. Pseudoviruses have been pivotal in conducting in vitro studies to understand the interaction between SARS-CoV-2 and host cells. They have helped in determining cell type susceptibility, virus receptors, and entry pathways for SARS-CoV-2. Hoffman and coauthors utilized pseudotyped SARS-CoV-2 particles to demonstrate that SARS-CoV-2 infection relies on host cell factors ACE2 and TMPRSS2. Indeed, it was reported that different cell lines exhibited varying degrees of susceptibility to SARS-CoV-2 entry, depending on a lower or higher expression of ACE2 receptor [43]. The pseudoviruses helped elucidate the role of host proteases, such as TMPRSS2 and Cathepsin B/L, in SARS-CoV-2 entry. It has been demonstrated that SARS-CoV-2 can use both Cathepsin B/L and TMPRSS2 for cell entry, particularly in human lung cells. Otherwise, the omicron variant of SARS-CoV-2 enters cells by endo/lysosomal cysteine protease Cathepsin L [44]. 

### 3.2. Neutralizing Antibody Assay

Pseudoviruses are widely used in virus neutralization assays to evaluate the efficiency of neutralizing antibodies. By measuring the ability of antibodies to block pseudovirus entry into target cells, it is possible to assess the potency of immune responses elicited by vaccines or natural infections. A modified HIV backbone vector was used to produce Nipah (NiV) pseudovirus and test, by in vitro and in vivo neutralization assay, the protective efficacy of antibodies specific against the virus [29]. Three groups of guinea pigs were immunized with plasmids expressing two of the outer membrane proteins of the Nipah virus (NiV), fusion protein (F) and glycoprotein (G) and both F and G. After immunization, sera containing neutralizing antibodies (NAbs), targeting specific antigens (F, G, or both F and G) were transferred into mice, which were subsequently inoculated with NiV pseudovirus. This approach allowed to the development of in vivo imaging mouse model for NiV pseudovirus and permitted data to be obtained regarding limited protection by individual NAbs, unlike full protection by using F/G NAbs. Similarly, the construction of the Chikungunya virus glycoprotein pseudotype (CHIKVpp) using lentiviral vectors represented a significant advancement in the study of CHIKV infection and immunity. Serum samples collected from three individuals infected with CHIKV were tested for their ability to block CHIKVpp infection. The results demonstrated that the addition of serum containing neutralizing antibodies from CHIKV-infected individuals effectively inhibited CHIKVpp infection in a concentration-dependent manner. This suggested that the neutralizing antibodies present in the serum were able to recognize and bind to the CHIKVpp, preventing entry into host cells and subsequent infection [45]. Antibody neutralization assays targeting S protein and based on SARS-CoV-2 pseudovirus have been carried out in multiple studies. Ferdinand Zettl and collaborators used pseudotyped VSV*ΔG(FLuc), a G-deleted VSV encoding both GFP and firefly luciferase, with the SARS-CoV-2 spike protein, to detect SARS-CoV-2 neutralizing antibodies in the sera of convalescent COVID-19 patients [10]. The in vivo studies conducted by Xi Wu and collaborators represented a significant advancement in understanding the protective efficacy of monoclonal antibodies against COVID-19. In particular, the authors developed a pseudovirus containing the firefly luciferase reporter gene able to be monitored in the infected tissues through in vivo bioluminescent imaging, and they evaluated the protective efficacy of BGB-DXP593, a monoclonal neutralizing antibody derived from the B cells of convalescent COVID-19 patients. Two doses of BGB-DXP593 were given to mice individually via the intravenous route, followed by intranasal rVSV-SARS-CoV-2. The administration of BGB-DXP593 resulted in complete protection against rVSV-SARS-CoV-2 highlighting its potential role as a therapeutic agent for COVID-19 [46]. 

### 3.3. Screening of Entry Inhibitors 

Pseudoviruses serve as a valuable tool for screening antiviral drugs and evaluating their ability to inhibit viral entry into host cells. This approach has accelerated the discovery and development of novel antiviral therapies, as demonstrated by various studies targeting different viral infections, including MERS-CoV and SARS-CoV-2. Moreover, in some cases, the pseudovirus technology was also used to test existing drugs with known safety profiles against emerging infections. Researchers evaluated the effectiveness of HIV entry inhibitors against MERS-CoV pseudovirus infection. They discovered that small molecules, ADS-J1, and 3-hydroxyphthalic anhydride-modified human serum albumin (HP-HSA), demonstrated significant inhibition of MERS-CoV pseudovirus infection [28]. Recently, HIV inhibitor molecules have been tested as inhibitors of SARS-CoV-2 variants. By pre-treating cells with these compounds, the authors investigated their ability to block viral entry by specifically inhibiting host proteases [47]. Moreover, in another study, a lentiviral-based pseudotyped particle of SARS-CoV-2 was used to assess the efficiency of membrane fusion inhibitors. IPB02, a lipopeptide fusion inhibitor, demonstrated high potency in inhibiting both SARS-CoV-2 and SARS-CoV pseudoviruses, while the unconjugated peptide, IPB01, showed weak inhibitory activity. This highlights the potential of IPB02 as a highly potent fusion inhibitor for both SARS-CoV-2 and SARS-CoV infections [48].

## 4. Natural Products Inhibiting Viral Entry

Natural products have garnered significant interest in recent years as potential sources of antiviral agents due to their diverse chemical structures and pharmacological activities. Pseudoviruses serve as valuable tools for screening and evaluating the antiviral potential of natural products, particularly in inhibiting viral entry into host cells. These compounds often target key viral proteins involved in the entry process, such as viral surface glycoproteins or host cell receptors, thereby preventing viral attachment, fusion, or entry into host cells. In this chapter, we report examples of several viral infections caused by highly pathogenic viruses, the underlying mechanisms of viral infection, and the significance of discovering alternative remedies to existing pharmacological therapies. Additionally, this chapter compiles a comprehensive overview of natural substances that have been investigated as potential antivirals against these viruses, by pseudovirus tools for screening purposes. 

### 4.1. Natural Therapeutic Compounds against Coronaviruses

Human coronaviruses were first isolated in the 1960s, belong to the Coronaviridae family, and causing respiratory infections in mammals, such as bats, camels, and civets [49]. Common strains of coronaviruses, including 229E, NL63, OC43, and HKU1, cause mild respiratory tract infections in humans worldwide [50]. Highly pathogenic strains were responsible for severe acute respiratory syndrome coronavirus (SARS-CoV) in 2002 and Middle East respiratory syndrome-CoV (MERS-CoV) in 2012. In December 2019, COVID-19 (coronavirus disease 2019) broke out in Wuhan, in the Hubei province of China, and rapidly spread across China and worldwide, causing a global pandemic [51]. This novel beta coronavirus, SARS-CoV-2, shares 79% of its genome sequence with SARS-CoV and around 50% with MERS-CoV [52]. The CoVs are membrane-enveloped, non-segmented, positive-strand RNA viruses. Both viruses share the same organizational structures, which include 16 non-structural proteins and an open reading frame (ORF) 1a/b located at the 5′ end. These are surrounded by the structural protein spike (S), envelope (E), and membrane (M). Additionally, another ORF at the 3′ end encodes a nucleocapsid (N). The S protein of SARS-CoV-2, which consists of the two subunits S1 and S2, plays an important role in viral entry by mediating receptor binding and membrane fusion [53]. The infection of host cells by SARS-CoV-2 begins with the attachment of the receptor-binding domain (RBD) of the S1 subunit to the host cell receptor ACE2. The spike protein of the virus needs to be activated by the TMPRSS2 protease, which is present on the host cell membrane. This activation occurs when the protease cleaves the spike protein at the S1/S2 and S2 sites, leading to conformational changes. These changes allow the virus to fuse with the host membrane and enter into the host cells. Despite clinical similarities between MERS and SARS, MERS-CoV uses dipeptidyl peptidase 4 (DPP4, also termed CD26) as a cellular receptor [54,55]. Conversely, SARS-CoV and SARS-CoV-2 enter host cells by binding spike protein to ACE2. SARS-CoV, MERS-CoV, and SARS-CoV-2 use cellular serine protease TMPRSS2 and endosomal cysteine proteases cathepsin B/L for spike protein priming. Consequently, the S protein is considered the most attractive target for SARS-CoV vaccines and therapeutic development [43,56,57,58,59].

Since high levels of ACE2 are associated with an increase in viral replication, the development of cell lines overexpressing ACE2 is a useful model for studying viral infection and new drugs. 

Secondary metabolites are small chemical molecules synthesized by plants that have been proven to have positive effects in an increasing number of studies [12]. Numerous active phytochemicals have been discovered to have an impact on viral replication, membrane permeability, and cellular processes [13,14,15]. Moreover, by utilizing distinct treatment conditions, it is possible to elucidate the mechanisms of action of antiviral compounds. In recent studies, two different treatments have been employed to differentiate the effects of potential antivirals directly on the cells or on the virus itself. The authors investigated the entry inhibitory activity of luteolin-7-O-glucuronide (L7OG) and folic acid (FA), against SARS-CoV-2 pseudotypes harboring alpha (α) and omicron (o) spike proteins. The cells were pretreated to block cellular receptors or proteases responsible for viral entry. Conversely, pseudovirus pretreatment was conducted to evaluate if the compounds could hamper viral entry directly by targeting the spike protein. The authors reported that both compounds, L7OG and FA, were effective in inhibiting the entry of both variants of SARS-CoV-2. However, L7OG displayed a higher level of entry inhibition against the alpha variant, while FA showed activity against both variants [60]. In addition, Ji Yeun Kim and coauthors identified three natural compounds, dihydrotanshinone, E-64-C, and E-64-D, which act against MERS-CoV by utilizing a pseudovirus that carries S protein from MERS-CoV [61]. Dihydrotanshinone is a lipophilic compound that is extracted from the root of *Salvia miltiorrhiza* Bunge, also known as “dansam” in Korean. This plant is commonly used in traditional Asian medicine. This plant contains also tanshinones, which showed a specific inhibitory activity against SARS-CoV 3CLpro and PLpro proteases [62]. The authors assessed the inhibitory effects of the compounds in pre- and post-attachment assays and found that the relative infectivity in the pre-attachment assays was significantly lower (≤50%) than in the post-attachment assays, indicating that the compounds block the entry step of the virus. *Ephedra sinica*, or “ma-huang” in the Chinese Name Pinyin System, has been extensively utilized in traditional Chinese medicine [63,64]. In China, it has also been used extensively to treat COVID-19 [65]. Ephedrine (EP), pseudoephedrine (PEP), and methylephedrine (MEP)) from the *Ephedra sinica* were able to bind with ACE2, preventing SARS-CoV-2 spike pseudovirus entry in ACE2h cells.

The pre-treatment of pseudovirus or target cells with *Spatholobus suberectus* Dunn (SSP) resulted in consistent inhibitory activity with the respective EC_50_ values of 2.3 or 2.1 μg/mL, respectively. SSP blocked both SARS-CoV-2 spike glycoprotein and the host receptor ACE2 [66]. Spirulina and green tea extracts blocked the entry of SARS, MERS, and SARS-2 pseudoviruses with high efficiency [67]. Cannabidiolic acid (CBDA) and cannabigerolic acid (CBGA) are two cannabinoids with a high affinity for the spike protein in viral neutralization assays. The infection of human epithelial cells with a pseudovirus expressing SARS-CoV-2 spike protein was inhibited by CBDA and CBGA, with an IC_50_ value of 7.7 μg/mL and 8.4 μg/mL, respectively.

Cannabidiolic acid (CBDA) and cannabigerolic acid (CBGA) are two cannabinoids with a high affinity for the spike protein in viral neutralization assays. The infection of human epithelial cells with a pseudovirus expressing SARS-CoV-2 spike protein was inhibited by CBDA and CBGA, with an IC50 value of 7.7 μg/mL and 8.4 μg/mL, respectively.

Additionally, it was also reported that cannabigerolic acid and cannabidiolic acid were equally effective against the SARS-CoV-2 alpha variant B.1.1.7 and beta variant B.1.351 [68]. Moreover, the epigallocatechin gallate (EGCG), 20(R)-ginsenoside Rg3 (RRg3), 20(S)-ginsenoside Rg3 (SRg3), isobavachalcone (Ibvc), isochlorogenic A (IscA) and bakuchiol (Bkc) were found to effectively inhibit pseudovirus entry at a concentration of 100 µM [69]. Some of these substances have previously been identified as SARS-CoV-2 antiviral drugs. In high-throughput screening studies, it was discovered that Bvc, Isl, and Ceph were effective against the SARS-CoV-2 virus [70,71]. The compounds were evaluated for their ability to prevent infection by a SARS-CoV-2-pseudotyped lentivirus encoding the EGFP reporter gene in HEK293 cells that were transiently co-overexpressing human ACE2 and mCherry. At doses up to 100 µM, 6 of the 7 compounds—EGCG, RRg3, SRg3, Ibvc, IscA, and Bkc—effectively prevented the entry of pseudoviruses, whereas SalA only moderately did so. Among these, EGCG and Ibvc prevented SARS-CoV-2 infection by inhibiting RBD and/or ACE2 [69]. Li Jun Yang and coauthors reported the effectiveness of corilagin from *Phyllanthus urinaria* towards SARS-CoV-2 by the construction of RBD-pseudotyped lentivirus. The authors reported a strong binding affinity of corilagin to SARS-CoV-2-RBD and hACE2 protein. Indeed, dose-dependent pretreatment with corilagin for 24 h blocked SARS-CoV-2-RBD binding and abolished the infectiousness of RBD-pseudotyped lentivirus in hACE2 overexpressing HEK293 cells [72]. The docking analysis of corilagin on SARS-CoV-2 was useful in predicting the preferential binding site of corilagin, which is located in a pocket containing the residues Cys 336 to Phe 374 of the receptor binding domain (RBD) of spike-RBD. *Sanguisorba officinalis* L., *Paeonia lactiflora* pall, and *Mangifera indica* represent some of the traditional medicinal herbs that naturally contain the polyphenolic compound known as 1,2,3,4,6-O-Pentagalloylglucose (PGG) [73,74,75]. PGG demonstrated a significantly higher binding affinity to SARS-CoV-2-RBD protein compared to hACE2. Additionally, PGG was able to eliminate the infectiousness of RBD-pseudotyped lentivirus in hACE2 overexpressing HEK293 cells [76]. Other mammalian and avian coronaviruses, as well as SARS-CoV, and MERS-CoV, were inhibited by lectin [77,78,79]. A pseudovirus-based neutralization assay was carried out by Wenbo Wang’s research groups to assess the anti-SARS-CoV-2 activity of 12 plant-derived lectins against mutant strains and epidemic variations [80]. The lentil lectin derived from *Lens culinaris* was able to successfully inhibit a range of mutant strains and variants, including synthetic mutants at the N-/O-linked glycosylation site, as well as the epidemic variants B.1.1.7, B.1.351, and P.1. This lectin specifically bound to oligomannose-type glycans and GlcNAc at the non-reducing end terminus, making it a promising avenue for further research into potential treatments for COVID-19. Moreover, coronaviruses require Ca^2+^ ions for host cell entry [81]. Thus, the identification of channel blockers (CCBs) could be a promising therapeutic option for the inhibition of viral entry. Chang-Long He and coauthors screened a library of 188 natural compounds using a luciferase-expressing pseudovirus encoding SARS-CoV-2 S (G614) protein. Nine compounds (SC9, SC161, SC171, SC182-187) with reasonably high activity (EC_50_ 10 μM), low cytotoxicity (CC_50_ > 20 μM), and high specificity (SI > 10, VSV-G EC_50_ > 20 μM) were chosen for in vitro inhibition assays in 293T-ACE2, Calu-3, and A549 transduced with coronaviruses (SARS-CoV, MERS-CoV, SARS-CoV-2 [S-D614, S-G614, and N501Y variants]). Except for SC171, all compounds were bis-benzylisoquinoline alkaloids. The substances were identified as channel blockers (CCBs) able to block the Ca^2+^ channels and prevent virus entry [71]. Yi L and coauthors screened a library containing extracts from 121 Chinese herbs and identified two molecules able to inhibit the internalization of wild-typed SARS-CoV-2 and HIV-luc/SARS pseudotyped virus [82]: TGG, a component of *Galla chinensis* and the luteolin, identified in various Chinese herbs like *Veronica linariifolia* Pall. Both compounds showed anti-SARS-CoV activity higher than that of glycyrrhizin, which was used as a positive antiviral control. Further, they discovered that quercetin, which is a structural analog of luteolin, had antiviral activity against HIV-luc/SARS, with an EC_50_ of 83.4 µM. Recently, a similar study was reported by Xiao, Z regarding SARS-CoV-2. A pseudovirus system tested the efficacy of quercetin and luteolin to inhibit SARS CoV-2 pseudovirus and genetic variants of SARS-CoV-2, including alpha variant B.1.1.7, beta variant B.1.351, delta variant B.1.617, and omicron [83].

Furthermore, using an HIV-1-based SARS-CoV-2 pseudotype, extracts from *Stachytarpheta cayennensis* and β-caryophyllene were found to inhibit SARS-CoV-2 entry into cells, among a library of 41 natural products including plant extracts, essential oils, and compounds from plants and microorganisms [84]. In particular, Stachytarpheta cayennensis showed an IC_50_ of 91.65 μg/mL, a CC_50_ of 693.5 μg/mL, and a selectivity index (SI) of 7.57, indicating its potential use as an inhibitor for SARS-CoV-2 entry. Similarly, an aqueous extract from the Natural herb *Prunella vulgaris* (NhPV) was able to interrupt the binding between the S glycoprotein of SARS-CoV-2 (including SPG614 mutant) to its receptor ACE2 [85], blocking the viral entry step. The results were also confirmed by infection with wild-type SARS-CoV-2 (hCoV-19/Canada/ON-VIDO-01/2020) virus in Vero cells [67] (Figure 2, Table 1).

### 4.2. Natural Products Inhibiting Viral Entry of HIV

HIV infection remains a significant global health challenge, with millions of people affected worldwide. One promising approach to fighting HIV is the usage of pseudoviruses to study and block HIV entry into target cells. Chang Rong Wang and collaborators demonstrated significant anti-HIV activity of methyl gallate (HEB) from the edible mushroom *Pholiota adiposa* [86]. They reported that HEB inhibited in a dose-dependent manner the viral entry of pseudovirus in TZM-BL cells with an IC_50_ value of 11.9 μM. Similarly, Mavhandu and collaborators generated an HIV-1 subtype C (HIV-1-C) pseudovirus to identify inhibitory substances from plants [87] by evaluating luciferase reporter gene expression. Catechin obtained from *Peltophorum africanum* inhibited HIV-1-C pseudovirus with selective indices of 6304 μM (IC_50_: 0.49 μM, CC_50_: 3089 μM). Conversely, crude methanol root extract of *Elaeodendron transvaalense* gave IC_50_ values of 11.11 μg/mL. Similarly, Liu and collaborators reported that *Spatholobus suberectus* Dunn percolation extract (SSP) has a broad spectrum of antiviral activity against the entry of SARS-CoV, H5N1, and HIV-1 [79]. The study included four anti-HIV drugs: AZT (zidovudine), a potent nucleoside reverse transcriptase inhibitor (NRTI); T20, an HIV-1 fusion blocker; Maraviroc (MVC), a CCR5 antagonist; and JM2987, a CXCR4 antagonist, which served as a positive control. The authors reported that the use of 50 μg/mL of SSP to treat the virus showed a significant inhibition of HIV-1 infection, similar to the effects of T-20. However, when target cells were pretreated with SSP, no significant viral inhibition was observed. These results imply that SSP hinders HIV-1 infection by targeting the viral envelope glycoprotein gp160, which is responsible for viral entry into host target cells (Figure 3, Table 2).

### 4.3. Natural Products Inhibiting Viral Entry of Influenza Virus

Influenza A virus (IAV) is a highly contagious respiratory pathogen belonging to the Orthomyxoviridae family and characterized by a segmented RNA genome, which allows for frequent genetic reassortment and the emergence of new strains with pandemic potential [88]. The RNA genome encodes for several functional, structural, and non-structural proteins, including PB1, PB2, PA, HA, NA, NP, M, and NS. Influenza A viruses are classified into subtypes based on the surface glycoproteins hemagglutinin (HA) and neuraminidase (NA), which play critical roles in viral entry and release from host cells, respectively [89]. Vaccination remains the primary strategy for preventing influenza A virus infection, with seasonal influenza vaccines typically updated annually to match circulating strains. In addition to vaccination, antiviral drugs such as neuraminidase inhibitors are used for treatment of and prophylaxis against influenza A virus infection. In this regard, recent studies have detailed the therapeutic impact of quercetin and its derivatives in combating the virus [90]. Researchers have identified 33 compounds in *Schinopsis brasiliensis*, such as corilagin, chlorogenic acid, and quercetin derivatives, with promising antiviral activities. In particular, Mehrbod et al. found, using pseudovirus-based inhibition assays, that quercetin inhibited the entry of the H5N1 virus [90]. Similar results were obtained by Wu and collaborators [91]. Overall, both studies provide valuable insights into the potential use of quercetin as a therapeutic option for the treatment and prophylaxis of IAV infections.

In another study, 15 Amaryllidaceae alkaloids isolated from the bulbs of L. radiata were tested for in vitro evaluation of antiviral activity against influenza virus type A, A/Chicken/GuangDong/178/2004 [92]. Viral entry inhibition assays using H5N1-HIV pseudoviruses demonstrated that the alkaloids did not inhibit the entry of H5N1 pseudoviruses, unlike SSP treatment on virus particles or target cells interferes with H5N1 pseudovirus internalization [66] (Figure 3, Table 3).

### 4.4. Natural Products Inhibiting Viral Entry of the Ebola Virus

EBOV (Ebola virus) is a contagious virus that belongs to the Filoviridae family and causes EVD (Ebola virus disease), previously known as EBOV hemorrhagic fever. The virus is spread to humans through contact with infected animals, such as fruit bats, primates, or forest antelope, or direct exposure to the bodily fluids of infected individuals [93]. EBOV has caused outbreaks of varying magnitude in Central and West Africa, with the largest outbreak occurring between 2014 and 2016 in West Africa, primarily affecting Guinea, Liberia, and Sierra Leone. Recent research has led to the development of experimental vaccines and treatments that have shown promising results in clinical trials. Pseudoviruses have played a crucial role in EBOV research, providing a safer and more controlled model for studying virus entry, replication, and pathogenesis. By using pseudoviruses bearing the EBOV glycoprotein (GP), researchers can investigate virus–host interactions, screen potential antiviral drugs, and evaluate the efficacy of candidate vaccines in preclinical studies.

The study by Cui and colleagues investigated a series of quinoline compounds for their potential as anti-EBOV entry inhibitors. Among these, compound SYL1712 emerged as the most potent inhibitor, with an IC_50_ of 1 μM and a SI > 200. Time-of-addition assays demonstrated that SYL1712 effectively blocks HIV-1/EBOV pseudovirus entry within one hour [94]. Shaikh and collaborators screened a database of natural compounds to identify those likely to interfere with GP in the attachment to host cells. HIV-1-derived pseudoviruses expressing Ebola virus envelope GPs were employed to discover the power of compounds ZINC32540717 and ZINC09410451 against EBOV infection in vitro [95]. Similarly, Zhang and collaborators reported that an aqueous extract from the Chinese herb *Prunella vulgaris* (CHPV) blocks the entry of an EBOV-GP pseudotyped [96]. Wang and colleagues conducted a study to assess the antiviral activity of teicoplanin, a glycopeptide antibiotic commonly used to treat severe bacterial infections, derived from *Actinoplanes teichomyceticus*. The authors performed a time-of-addition study using an EBOV pseudovirus model. They observed that teicoplanin effectively exerted its inhibitory effect specifically during the entry/attachment phase of the viral lifecycle, rather than inhibiting replication post-attachment. Thus, it inhibited pseudotyped EBOV infection in a dose-dependent manner [93] (Figure 3, Table 4).

### 4.5. Natural Products Inhibiting Viral Entry of Lassa Virus and Chikungunya Virus

Lassa virus and Chikungunya virus are important pathogens that pose public health concerns globally and require ongoing surveillance, research, and public health interventions to prevent and control outbreaks and mitigate their impact on affected communities.

Lassa virus (LASV) is an enveloped, single-stranded RNA virus belonging to the Arenaviridae family. It is primarily transmitted to humans through contact with infected rodents or their excretions. Lassa fever, caused by this virus, is endemic in parts of West Africa, particularly in countries like Nigeria, Liberia, and Sierra Leone. Currently, there are no authorized vaccines or antivirals for LASV (Lassa virus). Clinical treatment options are restricted to the use of the broad-spectrum antiviral ribavirin. Ke Tang and collaborators screened a library of 40 natural products from dietary supplements and evaluated their effects on LASV entry using the LASV-GP/HIV-luc pseudovirus [98]. The authors reported that capsaicin, naturally abundant in chili peppers, inhibited the entry of five LASV strains. Moreover, they studied the antiviral mechanism of capsaicin by performing a time-of-addition assay. The cells were incubated with capsaicin (30 μmol/L), either during LASV-GP/HIV attachment or post-attachment or throughout the entire process. The results reported that capsaicin blocked LASV entry by impeding LASV glycoprotein (GP)-mediated fusion [98].

Chikungunya virus (CHIKV) is an arthropod-borne virus belonging to the genus Alphavirus within the Togaviridae family. The transmission of CHIKV mainly occurs through Aedes mosquitoes, particularly Aedes aegypti and Aedes albopictus, which are known as arthropod-borne vectors.

The CHIKV genome consists of single-stranded positive-sense RNA and encodes non-structural and structural proteins. CHIKV infection typically presents with symptoms such as fever, rash, headache, muscle pain, and joint pain, which can be debilitating but is rarely fatal. Currently, although there is no specific antiviral treatment for CHIKV infection, various efforts are underway to develop effective vaccines and antiviral drugs. Epigallocatechin gallate (EGCG) is a natural compound found in green tea that has been studied for its potential antiviral properties against various viruses, including CHIKV. The antiviral effect of EGCG has been tested using a pseudoviruses screening assay, and the results reported that EGCG blocked the entry of CHIKV Env-pseudotyped lentiviral vectors, interfering with CHIKV attachment to target cells [99] (Figure 3, Table 5).

## 5. Discussion

The use of pseudoviruses as a research tool has revolutionized our understanding of viral infectivity, pathogenesis, and host–virus interactions. Pseudoviruses, being replication-defective, offer a safer alternative to studying highly pathogenic viruses, allowing researchers to manipulate them in biosafety level 2 facilities. Their construction involves various packaging systems, and each system offers distinct advantages and limitations. The HIV-1, MLV, and VSV packaging systems are commonly employed for pseudovirus construction. While HIV-1 and MLV systems are simpler and less time-consuming, VSV systems yield higher pseudovirus quantities despite being more complex. The choice of packaging system depends on experimental requirements, balancing factors like efficiency, scalability, and safety.

This manuscript provides a comprehensive overview of the construction, applications, and significance of pseudoviruses in virology research, particularly in screening and evaluating natural antiviral compounds.

Natural products have emerged as promising sources of antiviral agents due to their diverse chemical structures and pharmacological activities. By utilizing pseudovirus-based assays, researchers can screen and evaluate the efficacy of natural compounds in inhibiting viral entry into host cells. However, these assays are limited to viruses with specific envelope proteins, making them ineffective for viruses without envelope proteins, such as rotavirus and poliovirus.

In this review, we reported numerous studies that have identified phytochemical compounds that prevent viral attachment, fusion, or entry into host cells, targeting viral proteins involved in the entry process, such as viral surface glycoproteins or host cell receptors.

Among coronaviruses, SARS-CoV-2 has sparked significant research interest. Pseudovirus-based assays have facilitated the screening of natural compounds for their ability to inhibit SARS-CoV-2 entry into host cells. Furthermore, pseudovirus technology has been crucial in studying the mechanisms of action of potential antiviral compounds against SARS-CoV-2 and its variants [60,61,62,63,64,65,66,67,68,69,72,73,74,75,76,77,78,79,80,82,83,84,85]. Luteolin-7-O-glucuronide (L7OG) and folic acid (FA) showed promising entry inhibitory activity against SARS-CoV-2 pseudotypes harboring alpha and omicron spike proteins [60]. Similarly, dihydrotanshinone, E-64-C, and E-64-D were identified as effective against MERS-CoV by targeting the spike protein [61,62]. Compounds from *Ephedra sinica* (EP, PEP, MEP) inhibited SARS-CoV-2 spike pseudovirus entry by binding to ACE2 [63,64]. *Spatholobus suberectus* Dunn (SSP) demonstrated consistent inhibitory activity against SARS-CoV-2 by blocking spike glycoprotein and ACE2 [66]. Spirulina and green tea extract also effectively blocked the entry of SARS, MERS, and SARS-2 pseudoviruses [67]. Moreover, cannabidiolic acid (CBDA) and cannabigerolic acid (CBGA) inhibited SARS-CoV-2 spike pseudovirus entry, with similar effectiveness against alpha and beta variants as well as epigallocatechin gallate (EGCG), ginsenoside, and isobavachalcone [68,69].

Beyond coronaviruses, pseudovirus technology has been applied to study the entry mechanisms of other highly pathogenic viruses, including HIV, influenza, Ebola, Lassa, and Chikungunya. Wang and colleagues have shown that the edible mushroom *Pholiota adiposa* contains methyl gallate (HEB), which exhibits anti-HIV activity [86]. Similarly, the catechin identified from *Peltophorum africanum* and the crude methanol root extract of *Elaeodendron transvaalense* showed inhibitory activity against HIV-1 subtype C pseudovirus entry [87]. In another study, it was reported the broad-spectrum antiviral activity of *Spatholobus suberectus* Dunn percolation extract (SSP) against SARS-CoV, H5N1, and HIV-1 [66,79]. Studies conducted by Mehrbod and colleagues and Wu and team have suggested that quercetin could be effective in impeding the entry of the H5N1 virus, offering a promising therapeutic approach against viral infections [90,91]. For EBOV, within quinoline compounds, it was identified SYL1712 as a potent inhibitor of EBOV entry [94], as well as it was found that ZINC32540717 and ZINC09410451 interfere with the attachment to host cells [95]. Zhang and colleagues reported that an aqueous extract from *Prunella vulgaris* inhibited EBOV-GP pseudotyped HIV-based vector infection [96]. Additionally, was observed that the teicoplanin inhibits EBOV pseudovirus infection during the entry/attachment phase [97].

Regarding LASV, Tang and colleagues screened natural products and found that capsaicin inhibits LASV entry by impeding LASV glycoprotein-mediated fusion [98]. For CHIKV, EGCG showed potential antiviral properties by blocking CHIKV attachment to target cells [99].

These studies have led to the identification of natural compounds with potent antiviral activity against a range of viral infections. In addition to pseudovirus tools, minigenomes and replication-competent virus-like particles (trVLPs) provide a safe method for studying highly pathogenic viruses. Minigenomes are modified versions of viral genomes in which viral genes are replaced by reporter genes, such as firefly luciferase or eGFP, flanked by necessary elements for replication and transcription. These minigenomes cannot produce infectious viruses, as they lack essential viral proteins, making them valuable for screening antiviral compounds targeting viral replication. One model commonly used is the Ebola virus (EBOV) minigenome [96]. An extension of minigenome systems is represented by trVLPs, which include a minigenome along with the ribonucleoprotein required for replication and transcription, as well as envelope proteins [100]. This allows trVLPs to simulate the viral particle’s route during infection in a safe laboratory setting.

Combining pseudovirus and minigenome tools enables a comprehensive examination of viral inhibition mechanisms [97]. This integrated approach enhances our understanding of how antiviral compounds interfere with viral replication and provides valuable insights for developing antiviral therapies.

This review provides insights into various highly pathogenic viruses, elucidating their mechanisms of infection and the importance of discovering alternative antiviral therapies. Although they are not intended to replace pharmacological therapies, natural products provide valuable support and prevention where the virus is endemic.

## Figures and Tables

**Figure 1 ijms-25-05188-f001:**
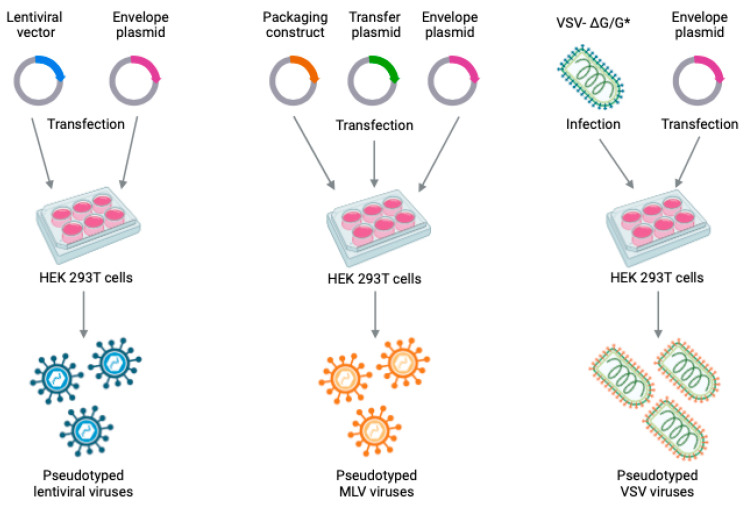
Representation of three different packaging systems for pseudovirus production (reviewed from the paper by Xiang Q et al., 2022 [2]).

**Figure 2 ijms-25-05188-f002:**
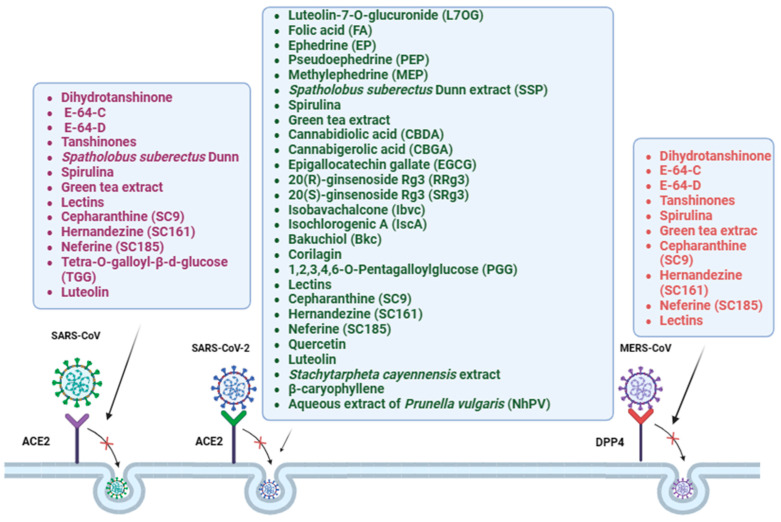
Natural Therapeutic Compounds against Coronaviruses. Graphical representation of natural compounds that have shown potential therapeutic activity against SARS-CoV, MERS-CoV, and SARS-CoV-2.

**Figure 3 ijms-25-05188-f003:**
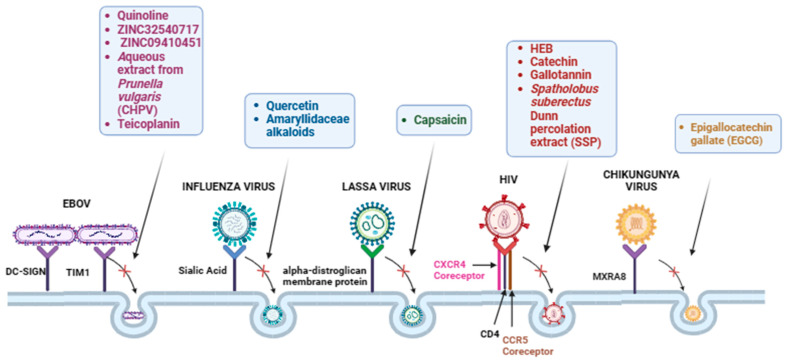
Natural Therapeutic Compounds against Viruses. Graphical representation of natural compounds that have shown potential therapeutic activity against EBOV, influenza virus, Lassa virus, HIV, and Chikungunya virus.

**Table 1 ijms-25-05188-t001:** Natural substances active against coronaviruses.

Substance	Target	Reference
Luteolin-7-O-glucuronide (L7OG) and folic acid (FA)	SARS-CoV-2	[60]
Dihydrotanshinone E-64-C, E-64-D, Tanshinones	SARS-CoV	[61,62]
	MERS-CoV	
Ephedrine, pseudoephedrine, and methylephedrine	SARS-CoV-2	[63,64]
*Spatholobus suberectus* Dunn	SARS-CoV-1 SARS-CoV-2	[66]
Spirulina and green tea extract	SARS-CoV-1 SARS-CoV-2 MERS-CoV	[67]
Cannabidiolic acid and cannabigerolic acid	SARS-CoV-2	[68]
Epigallocatechin gallate (EGCG), 20(R)-ginsenoside Rg3 (RRg3), 20(S)-ginsenoside Rg3 (SRg3), isobavachalcone (Ibvc), isochlorogenic A (IscA), and bakuchiol (Bkc)	SARS-CoV-2	[69]
*Phyllanthus urinaria* corilagin	SARS-CoV-2	[72]
1,2,3,4,6-O-Pentagalloylglucose (PGG)	SARS-CoV-2	[73,74,75,76]
Lectins	SARS-CoV-2 SARS-CoV-1 MERS-CoV	[77,78,79,80]
SC9, SC161, SC182-187	SARS-CoV-2 SARS-CoV-1 MERS-CoV	[71]
Tetra-O-galloyl-β-d-glucose (TGG) and luteolin	SARS-CoV	[82]
Quercetin and luteolin	SARS-CoV-2	[83]
Extract of *Stachytarpheta cayennensis* and β-caryophyllene	SARS-CoV-2	[84]
Aqueous extract of *Prunella vulgaris*	SARS-CoV-2	[85]

**Table 2 ijms-25-05188-t002:** Natural substances active against HIV.

Substance	Target	Reference
Methyl gallate (HEB)	HIV	[86]
Catechin Crude methanol root extract of *Elaeodendron transvaalense*	HIV	[87]
*Spatholobus suberectus* Dunn percolation extract (SSP)	HIV	[79]

**Table 3 ijms-25-05188-t003:** Natural substances active against Influenza virus.

Substance	Target	Reference
Quercetin	H5N1	[90,91]
Amaryllidaceae alkaloids	H5N1	[92]
*Spatholobus suberectus* Dunn percolation extract (SSP)	H5N1	[66]

**Table 4 ijms-25-05188-t004:** Natural substances active against Ebola virus.

Substance	Target	Reference
SYL1712	EBOV	[94]
ZINC32540717 and ZINC09410451	EBOV	[95]
Aqueous extract from the Chinese herb *Prunella vulgaris* (CHPV)	EBOV	[96]
Teicoplanin	EBOV	[97]

**Table 5 ijms-25-05188-t005:** Natural substances active against the Lassa virus and Chikungunya virus.

Substance	Target	Reference
Capsaicin	Lassa virus	[98]
Epigallocatechin gallate (EGCG)	Chikungunya virus	[99]

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
