# Peer review of "Pseudovirus-Based Systems for Screening Natural Antiviral Agents: A Comprehensive Review"

_ijms, 2024, doi:10.3390/ijms25105188_

Round 1

Reviewer 1 Report

Comments and Suggestions for Authors

The review entitled ‘’ Pseudovirus-Based Systems for Screening Natural Antiviral Agents: A Comprehensive Review’’ provides a detailed summary on the significance of pseudoviruses along with their three packaging and production systems. The authors have given insights regarding the mechanism of infection of various disease causing viruses, thereby analyzing several natural products based antiviral agents against pseudoviruses.

u  The figure 1 is slightly modified version of figure provided in earlier published paper (doi : https://doi.org/10.1016/j.micres.2022.126993), cite corresponding reference with the figure.

u  All the figures should be provided in high resolution to avoid blurry vision. Moreover, in figure 2 & 3, names of several natural products based antiviral agents may be presented with bullets (with different color) for quick understanding.

u  As for category 4.1, a separate table should also be given for each category to summarize the natural products inhibiting the viral entry of each discussed virus.

u  Though the provided discussion is detailed, however, it lacks the description of summarized work regarding the utilization of various natural products to inhibit the viral entry of different viruses.

u  Overall, the references are given appropriately. However, in few references, year is not bold, which must be carefully checked and revised.

  Minor Remarks:

o   Line 20 & 21 need to be revised.

o   Line 26, spelling of ‘’against’’ need to be corrected. Also avoid extra spacing.

o   Line 36-38 & 192 should be revised to avoid grammatical mistakes.

o   Line 38, fordeveloping? Add space.

o   Line 146 & 147 are not implying clear meaning. Revise.

o   EC50 and IC50 should be written in a standard form throughout the manuscript.

o   Line 363, ‘’Green tea extracts’’, why first letter is being capitalized?

Comments on the Quality of English Language

Numerous sentence structure related mistakes are present in the manuscript. As a result, many sentences fail to give the clarity. A thorough and critical proof-reading of manuscript is suggested to avoid the existing multiple grammatical mistakes.

Author Response

We sincerely thank the Reviewer for the comments, which were a great help in revising the manuscript. In our opinion, this considerably improved the first version of the manuscript. Please find below a point-by-point description that includes the original Reviewer’s comments in boldface and the responses in italic typeface. We specify that we resubmit the new version of the manuscript with several modifications compared to the previous version.

The review entitled ‘’ Pseudovirus-Based Systems for Screening Natural Antiviral Agents: A Comprehensive Review’’ provides a detailed summary on the significance of pseudoviruses along with their three packaging and production systems. The authors have given insights regarding the mechanism of infection of various disease causing viruses, thereby analyzing several natural products based antiviral agents against pseudoviruses.

  1. The figure 1 is slightly modified version of figure provided in earlier published paper (doi : https://doi.org/10.1016/j.micres.2022.126993), cite corresponding reference with the figure.

Thanks to the reviewer comment. We have modified the figure accordingly and added the relative reference.

  1. All the figures should be provided in high resolution to avoid blurry vision. Moreover, in figure 2 & 3, names of several natural products based antiviral agents may be presented with bullets (with different color) for quick understanding.

We apologise for the incovinience, please new figures in high resolution have been uploaded for the new version of the manuscript.

  1. As for category 4.1, a separate table should also be given for each category to summarize the natural products inhibiting the viral entry of each discussed virus.

Thanks to the reviewer suggestion, we added the table accordingly.

  1. Though the provided discussion is detailed, however, it lacks the description of summarized work regarding the utilization of various natural products to inhibit the viral entry of different viruses.

As suggested by the reviewer, we added a description accordingly.

  1. Overall, the references are given appropriately. However, in few references, year is not bold, which must be carefully checked and revised.

Sorry for the mistake. Please see the new version of the bibliography.

  Minor Remarks:

  • Line 20 & 21 need to be revised.
  • Line 26, spelling of ‘’against’’ need to be corrected. Also avoid extra spacing.
  • Line 36-38 & 192 should be revised to avoid grammatical mistakes.
  • Line 38, fordeveloping? Add space.
  • Line 146 & 147 are not implying clear meaning. Revise.
  • EC50and IC50 should be written in a standard form throughout the manuscript.

-Line 363, ‘’Green tea extracts’’, why first letter is being capitalized?

As suggested by the reviewer, we resolved the minor remarks as reported in a new version of our manuscript.

Comments on the Quality of English Language

-Numerous sentence structure related mistakes are present in the manuscript. As a result, many sentences fail to give the clarity. A thorough and critical proof-reading of manuscript is suggested to avoid the existing multiple grammatical mistakes.

We appreciate your feedback. It's important to note that the manuscript underwent supervision and thorough editing by linguist experts, who are also co-authors of the paper.

Reviewer 2 Report

Comments and Suggestions for Authors

The review entitled “Pseudovirus-Based Systems for Screening Natural Antiviral Agents: A Comprehensive Review” (Manuscript ID: ijms-2960797) describes the potential of pseudovirus-based techniques as versatile tools for identifyingnew viral entry inhibitors, providing an overview of packaging systems commonly employed for the construction of pseudoviruses and focusing on their use for the screening and evaluation of natural products as potential antiviral agents.

Overall, the manuscript provides a sufficiently clear explanation of the topic. However, there are some issues that should be addressed before publication.

• Page 2, lines 45-47: the sentence “pseudovirus-based tools can be safely utilized in biosafety level 2 facilities to study the internalization of highly infectious viruses and other aspects of viral infectivity without the need for high-level containment measures” sound like an unnecessary repetition of what authors said in lines 36-38 (“pseudoviruses can be easily manipulated in biosafety level 2 laboratories, to investigate the internalization mechanism adopted by viruses, study cellular tropism, individuate cellular receptors”), so please try to rephrase it.

• Page 2, line 55: in the introduction section, authors mightinclude a few examples of antiviral agents of each class identified so far, depending on their mechanism of action.Since reference 16 is outdated, it should be removed and the following references should be added: [J. Membrane Biol.2020, 253, 425–444. doi: 10.1007/s00232-020-00136-z] [International Journal of Immunopathology and Pharmacology 2021, 35. doi: 10.1177/20587384211002621] [European Journal of Medicinal Chemistry 2023, 249,115136. doi: 10.1016/j.ejmech.2023.115136].

• Page 2, line 58: please rephrase the title of the paragraph to avoid the repetition of the word “production”. A better alternative could be: “Packaging systems and strategies for pseudoviruses production”.

• Page 2, lines 59-72: this paragraph describes three packaging systems commonly used for the construction of pseudoviruses, providing an overview of advantages and disadvantages of each method. However, they are reported in a very confusing way and there are no conclusions that can be useful to guide the reader in choosing the best method. Therefore, in my opinion, this paragraph should be rephrased to make the authors' point of view clearer.

Other minor points are listed below:

• Page 2, line 52: please remove the hyphen (“Anti-viral”) and also the typos on pag. 1, line 26 (“activea gainst” for “active against”) and line 38 (“fordeveloping” for “fordeveloping”), as well as on pag. 10, line 426 (“SARSCoV- 2” and “SARS CoV- 2” for “SARS-CoV-2”) and on pag. 13, line 553 (“nonstructural” for “non-structural”).

• Page 3, line 122: “in vivo” should be in italics.

• Page 5, line 193: according to style of the manuscript, the title of the subparagraph 3.1 should be in italics.

• Page 6, line 217: “in vitro” should be in italics.

• Page 7, line 317: please replace “the receptor-binding domain (RBD) of the S1 protein” with “the receptor-binding domain (RBD) of the S1 subunit”.

• Page 9, line 424: the EC50 value is probably incorrect (83.4 M).

• Page 14, line 563: The quality of Figure 3 should be improved.

• Please replace, throughout the manuscript, “EC50”, “IC50” and “CC50” with “EC50”, “IC50” and “CC50”.

Finally, extensive editing of English language is requested since most of the sentences are not well structured, highlighting a poorknowledge of basic grammar. Furthermore, in some points of the manuscript the meaning of the sentences is not easily accessible. Just a few examples are reported below:

− Page 4, lines 152-153: “High titer pseudoviruses was obtained when these three plasmids were co-transfected into the cells, high titer pseudoviruses was obtained”.

− Page 6, lines 251-252: “Various studies have performed antibody neutralization assays targeting S protein and based on SARS-CoV-2 pseudovirus”.

− Page 6, lines 261-262: “Mice were administered with twodoses of BGB-DXP593 individually via the intravenous route”.

− Page 8, lines 346-347: “Ji Yeun Kim and coauthors have been identified three natural compounds, dihydrotanshinone, E-64-C, and E-64-D, which act against MERS-CoV”.

− Page 11, lines 465-466: “Unlike by pre-treating target cells with SSP, the authors did not have a significant antiviral effect”.

For these reasons, in my opinion, the manuscript is suitable for publishing in International Journal of Molecular Sciences after major revisions.

Comments on the Quality of English Language

Extensive editing of English language required

Author Response

We sincerely thank the Reviewer for the comments, which were a great help in revising the manuscript. In our opinion, this considerably improved the first version of the manuscript. Please find below a point-by-point description that includes the original Reviewer’s comments in boldface and the responses in italic typeface. We specify that we resubmit the new version of the manuscript with several modifications compared to the previous version.

The review entitled “Pseudovirus-Based Systems for Screening Natural Antiviral Agents: A Comprehensive Review” (Manuscript ID: ijms-2960797) describes the potential of pseudovirus-based techniques as versatile tools for identifyingnew viral entry inhibitors, providing an overview of packaging systems commonly employed for the construction of pseudoviruses and focusing on their use for the screening and evaluation of natural products as potential antiviral agents.

Overall, the manuscript provides a sufficiently clear explanation of the topic. However, there are some issues that should be addressed before publication.

  1. Page 2, lines 45-47: the sentence “pseudovirus-based tools can be safely utilized in biosafety level 2 facilities to study the internalization of highly infectious viruses and other aspects of viral infectivity without the need for high-level containment measures” sound like an unnecessary repetition of what authors said in lines 36-38 (“pseudoviruses can be easily manipulated in biosafety level 2 laboratories, to investigate the internalization mechanism adopted by viruses, study cellular tropism, individuate cellular receptors”), so please try to rephrase it.

Thanks for your comment. We have extensively modified te paragraph.

  1. Page 2, line 55: in the introduction section, authors mightinclude a few examples of antiviral agents of each class identified so far, depending on their mechanism of action.Since reference 16 is outdated, it should be removed and the following references should be added: [Membrane Biol.2020, 253, 425–444. doi: 10.1007/s00232-020-00136-z] [International Journal of Immunopathology and Pharmacoogy 2021, 35. doi: 10.1177/20587384211002621] [European Journal of Medicinal Chemistry 2023, 249,115136. doi: 10.1016/j.ejmech.2023.115136]

Thanks for your comment. We have replaced the reference 16 with those suggested by the reviewer.

  1. Page 2, line 58: please rephrase the title of the paragraph to avoid the repetition of the word “production”. A better alternative could be: “Packaging systems and strategies for pseudoviruses production”.

We are grateful to the reviewer suggestion. We have modified the text accordingly.

  1. Page 2, lines 59-72: this paragraph describes three packaging systems commonly used for the construction of pseudoviruses, providing an overview of advantages and disadvantages of each method. However, they are reported in a very confusing way and there are no conclusions that can be useful to guide the reader in choosing the best method. Therefore, in my opinion, this paragraph should be rephrased to make the authors' point of view clearer.

We have modified the paragraph taking into account the reviewer comment.

Other minor points are listed below:

-Page 2, line 52: please remove the hyphen (“Anti-viral”) and also the typos on pag. 1, line 26 (“activea gainst” for “active against”) and line 38 (“fordeveloping” for “fordeveloping”), as well as on pag. 10, line 426 (“SARSCoV- 2” and “SARS CoV- 2” for “SARS-CoV-2”) and on pag. 13, line 553 (“nonstructural” for “non-structural”).

-Page 3, line 122: “in vivo” should be in italics.

-Page 5, line 193: according to style of the manuscript, the title of the subparagraph 3.1 should be in italics.

-Page 6, line 217: “in vitro” should be in italics.

-Page 7, line 317: please replace “the receptor-binding domain (RBD) of the S1 protein” with “the receptor-binding domain (RBD) of the S1 subunit”.

-Page 9, line 424: the EC50 value is probably incorrect (83.4 M).

-Page 14, line 563: The quality of Figure 3 should be improved.

-Please replace, throughout the manuscript, “EC50”, “IC50” and “CC50” with “EC50”, “IC50” and “CC50”.

Finally, extensive editing of English language is requested since most of the sentences are not well structured, highlighting a poorknowledge of basic grammar. Furthermore, in some points of the manuscript the meaning of the sentences is not easily accessible. Just a few examples are reported below:

− Page 4, lines 152-153: “High titer pseudoviruses was obtained when these three plasmids were co-transfected into the cells, high titer pseudoviruses was obtained”.

− Page 6, lines 251-252: “Various studies have performed antibody neutralization assays targeting S protein and based on SARS-CoV-2 pseudovirus”.

− Page 6, lines 261-262: “Mice were administered with twodoses of BGB-DXP593 individually via the intravenous route”.

− Page 8, lines 346-347: “Ji Yeun Kim and coauthors have been identified three natural compounds, dihydrotanshinone, E-64-C, and E-64-D, which act against MERS-CoV”.

− Page 11, lines 465-466: “Unlike by pre-treating target cells with SSP, the authors did not have a significant antiviral effect”.

As suggested by the reviewer, we resolved the minor remarks as reported in a new version of our manuscript.

Comments on the Quality of English Language: Extensive editing of English language required

We appreciate your feedback. It's important to note that the manuscript underwent supervision and thorough editing by linguist experts, who are also co-authors of the paper.

Round 2

Reviewer 1 Report

Comments and Suggestions for Authors

The revised version of manuscript entitled ‘’ Pseudovirus-Based Systems for Screening Natural Antiviral Agents: A Comprehensive Review’’ doesn't comprehensively address the recommended modifications. For example,

a. Figures 2 and 3 still need modifications as per previous suggestion.

b. The "Discussion" section is not up to the mark and has not been revised.

c. Most importantly, the plagiarism of the entire manuscript is 40%, which is too high to consider this article for publication.

Comments on the Quality of English Language

Careful proof reading of manuscript is highly recommended to remove grammatical errors and typos.

Author Response

We sincerely thank the Reviewer for the comments, which were a great help in revising the manuscript. In our opinion, this considerably improved the first version of the manuscript. Please find below a point-by-point description that includes the original Reviewer’s comments in boldface and the responses in italic typeface. We specify that we resubmit the new version of the manuscript with several modifications compared to the previous version.

The revised version of manuscript entitled ‘’ Pseudovirus-Based Systems for Screening Natural Antiviral Agents: A Comprehensive Review’’ doesn't comprehensively address the recommended modifications. For example,

  1. Figures 2 and 3 still need modifications as per previous suggestion.

We thank the reviewer for the observation. For quick understanding, we revisited both figures 2 and 3 by changing the color of the box related to viral infection.

  1. The "Discussion" section is not up to the mark and has not been revised.

We are grateful to the reviewer for this suggestion. We edited the Discussion as per the reviewer’s suggestions.

  1. Most importantly, the plagiarism of the entire manuscript is 40%, which is too high to consider this article for publication.

We are grateful to the reviewer for this information. We checked the plagiarism percentage before uploading the file and we did not get a 40% percentage. Anyway, we edited the manuscript according to the detection file used for the initial assessment that was provided to us by the editor.

Reviewer 2 Report

Comments and Suggestions for Authors

Authors made the required changes. The article is now suitable for publication.

Author Response

We sincerely thank the Reviewer for the comments, which were a great help in revising the manuscript. 

Round 3

Reviewer 1 Report

Comments and Suggestions for Authors

The authors have comprehensively revised the manuscript and can now be accepted for publication.